# The time to initiate trophic feeding and its predictors among preterm neonate admitted to neonatal intensive care unit, Multicenter study, Northwest Ethiopia

**Daniel Adimasu Kebede[1], Yilikal Tafere[2], Tewodros Eshete[2], Ermias Abebaw[3]\*, Mekonen Adimasu[4], Bekalu Endalew[2]**

**1** Debre Markos Comprehensive Specialized Hospital, Debre Markos, Ethiopia, **2** Department of Public Health, College of Health Sciences, Debre Markos University, Debre Markos, Ethiopia, **3** School of Medicine, Debre Markos University, Debre Markos, Ethiopia, **4** Department of Nursing, School of Nursing and Midwifery, Addis Ababa University, Addis Ababa, Ethiopia

\* fjeremy23@gmail.com

## Abstract

### Background

Trophic feeding is a small volume, hypo-caloric feeding, gut priming or minimal enteral feeding acclimate the immature gut of enteral fasting preterm neonates. Delayed starting of trophic feeding had resulted in short and long-term physical and neurological sequels. The current study aimed to estimate the time to initiate trophic feeding and its predictors among preterm neonates admitted in the neonatal intensive care unit of Debre Markos, Felege Hiwot, and Tibebe Ghion comprehensive specialized hospitals.

### Methods

An institutional-based prospective follow-up study was conducted among 210 neonates. The data were collected with interview and chart review, entered into Epi data 3.1 and exported to Stata 14.1 for analysis. Multivariable Cox regression models were fitted to identify predictors of time to initiate trophic feeding.

### Result

A total of 210 neonates were followed for 10136 person-hours of risk time and 191 (90.95%) of neonates were started trophic feeding. The overall incidence of starting trophic feeding was 2 per 100 (95% CI: 2, 2.2) person-hours observations. The median survival time was 42 hours (95% CI: 36, 48). APGAR- score at first minute <7 (AHR: 0.6, 95% CI: 0.44, 0.82), gestational age of <34 weeks (AHR: 0.69, 95% CI: 0.5, 0.94), presence of respiratory distress syndrome (AHR: 0.5, 95% CI: 0.36, 0.68), presence of hemodynamic instability (AHR: 0.37, 95% CI: 0.24, 0.57), presence of perinatal asphyxia (AHR: 0.63, 95% CI: 0.44, 0.89), cesarean section delivery (AHR: 0.63, 95% CI: 0.44, 89) and being delivered within the

**Data Availability Statement:** All relevant data are within the paper.

**Funding:** The author(s) received no specific funding for this work.

**Competing interests:** The authors have declared that no competing interests exist.

**Abbreviations:** AGA, Appropriate for Gestational Age; AHR, Adjusted Hazard Ratio; APGAR, Appearance, Pulse, Grammies, Activity, Respiration; BW, Birth Weight; CHR, Crude Hazard Ratio; CI, Confidence Interval; CPAP, Continuous Positive Airway Pressure; DMCSH, Debre Markos Comprehensive Specialized Hospital; ELBW, Extremely Low Birth Weight; EN, Enteral Nutrition; EF, Enteral Feeding; EFY, Ethiopian Fiscal Year; FHCSH, Felege Hiwot Comprehensive Specialized Hospital; GA, Gestational Age; HR, Hazard Ratio; IQR, Interquartile Range; IUGR, Intrauterine Growth Restriction; KMC, Kangaroo Mother Care; LBW, Low Birth Weight; LGA, Large For Gestational Age; MEN, Minimal Enteral Nutrition; NEC, Necrotizing Enterocolitis; NICU, Neonatal Intensive Care Unit; PPH, Postpartum Hemorrhage; NPO, Nothing Per Os; SD, Standard Deviation; SGA, Small for Gestational Age; TGCSH, Tibebe Ghion Comprehensive Specialized Hospital; SPSS, Statistical Package for Social Sciences.

study hospitals (AHR: 0.54, 95% CI: 0.39, 0.74) were found to be statistically significant predictors of time to initiate trophic feeding.

## Conclusion

There was a significant delay to initiate trophic feeding in the studied hospitals. Gestational age of below 34 weeks, APGAR-score of less than seven, out-born delivery, cesarean delivery, presence of respiratory distress syndrome; perinatal asphyxia, and hemodynamic instability were predictors of delay in starting of trophic feeding. Standardized feeding guideline has to be implemented to overcome delays in enteral feeding initiation.

## Background

Preterm birth is defined as a birth before 37 completed weeks of gestation or fewer than 259 days from the first date of a woman's last menstrual period. The global estimate of preterm birth was 11.1% and 10.6% in 2010 and 2014, respectively. This has a great regional variation which was ranged approximately 5% in some European countries; and 18% in Sub-Saharan countries. This showed that low and middle-income countries account for the majority of the world's preterm births, 60% of it occurred in Sub-Saharan and South Asia countries [1,2].

Provision of intense nutritional support with both parenteral and enteral nutrition to preterm neonates born prematurely is necessary. This helps to attain the rate and composition of weight gain approximating the normal neonate at the postmenstrual age, to minimize postnatal growth failure and the risk of enterocolitis; and to optimize neurodevelopmental and long term health outcomes. To tackle this problem trophic feeding (TF) is the preferred choice of preterm feeding [3].

Trophic feeding is usually defined as serving small volumes of enteral feeding (EF) which is started within the first few postnatal days. It is minimal enteral nutrition (MEN), gut priming, or hypo-caloric feeding that acclimate the immature gut of enteral fasting in preterm neonates. It is feeding nutritionally insignificant amounts (1–2 mL/kg/dose or 10-15ml/kg/day) for immature neonates but stimulates and supplies nutrients to the developing gastrointestinal system [3]. Its aim is not to feed the baby but the intestine, preferably with colostrum [4]. It is the nutrition provided shortly after birth in an attempt to avoid or reduce parenteral nutrition and its related complications [5–8].

Those starting TF have more energy intake, improved feeding tolerance, greater and faster weight gain and head growth, less sepsis, significantly fewer days of parenteral nutrition and oxygen supplementation, and consequently earlier discharge [9–14].

TF reduces the incidence of complications related to enteral fasting and parenteral nutrition like hyper-bilirubinemia, gut atrophy, decrease gastrointestinal mass, infections, and cholestasis [13,15–20], and metabolic complications [16] without an increase in the risk of NEC [13]. Early initiation of EF with the mother's own milk and prevention of postnatal growth failure is the target of nutrition in preterm neonates [4,6].

Researches demonstrated that there is a considerable delay in the initiation of enteral feeding in preterm neonates worldwide. For instance, researchers reported that 80 to 90% of neonates did not start TF until 48 hours of age. And in another study, 40% of neonates were in enteral fasting until 96 hours of age [21,22]. According to one study 76% of preterm neonates had been kept nothing per os (NPO) during the first 24 hours and 22% were kept for 72 hours.

These neonates who started enteral feeding was getting only 10% dextrose intravenously. Overall, the neonates were kept NPO in 26.8% of the time while they were in NICU [23].

Despite the general recommendation to initiate early enteral feeding, a considerable number of the preterm neonates are kept NPO in the first few days and receiving only maintenance fluid. The inherent problems of immature gut motility and function as well as the fear of necrotizing enterocolitis and feeding intolerance are the two major reasons that delay the start of preterm feeding [13]. This is associated with an increased risk of neonatal mortality [23]. There is a marked dose response of increasing risk of neonatal mortality with increasing delay in initiation of breastfeeding. The overall late initiation was associated with a 2.4-fold increase in the risk of neonatal mortality [24] and other neonatal bad outcomes [23,25] and highly associated with postnatal growth failure [26].

The initiation, mode, and rate of advancement of TF remained a topic of argument [20,27]. Practices across NICUs and professionals are different. This variability includes preterm feeding practice which is not in line with the national guidelines [23].

Different findings indicate that there is a significant delay in starting TF across the globe and evidence on time to TF among preterm neonates admitted to NICU in Ethiopia is not adequately searched. Therefore, this study was aimed to estimate the time to initiate TF among preterm neonates admitted to NICU in three comprehensive specialized hospitals found in Debre Markos and Bahir Dar, Northwest Ethiopia.

## Methods

### Study design and settings

An institutional-based prospective follow up study was conducted. The study was conducted in neonatal intensive care units of Debre Markos, Felege Hiwot, and Tibebe Ghion comprehensive specialized hospitals, that the former found in Debre Markos town and the latter two found in Bahir Dar town, Amhara.

Debre Markos is located 300km from Addis Ababa, the capital city of Ethiopia, and 256 km from Bahir-Dar, which is 556 km away from Addis Ababa. Debre Markos (DMCSH), Felege Hiwot (FHCSH), and Tibebe Ghion (TGCSH) Comprehensive Specialized Hospitals have 162, 230, and 151 monthly and 1845, 2479, and 1652 annual neonatal admission according to the 2012 EFY report, respectively. Of these 63, 89, and 67; and 612, 834, and 660 were monthly and annual admissions of preterm neonates, respectively. These hospitals are equipped with 28, 43, and 27 neonatal beds and 24, 35, and 18 maternal beds respectively.

All of these hospitals are the final referral choice for other health institutions around and provide tertiary level neonatal care and are organized with necessary materials, equipment and health care workers with different professions (nurses, physicians, laboratory and pharmacist).

These are organized into different service areas including term, preterm, isolation, and procedure rooms with kangaroo mother care (KMC) and maternal waiting rooms. The major services are general neonatal care, blood and exchange transfusion, phototherapy, and ventilation support such as continuous positive airway pressure (CPAP). This study was conducted in these hospitals from October 1 to November 30, 2020.

The study population included all preterm neonates admitted to the neonatal intensive care unit of Debre Markos, Felege Hiwot, and Tibebe Ghion comprehensive specialized hospitals during the study period. Neonates born after 28 completed weeks, but before 37 completed weeks, and admitted to the neonatal intensive care unit of these hospitals were included in the study while Neonates who had started direct breast milk or other option of feeding before the time of admission either at home or referred facilities, pre-diagnosed stage II/III necrotizing enter colitis, stage III perinatal asphyxia, unknown gestational age, unknown APGAR score,

and birth weight, neonates who had started TF before admission were excluded from this study.

## Sample size and sampling techniques

The sample size was computed by using STATA (version 14) considering this statistical assumptions; two-sided significant level ($\alpha$) of 5%, power 80%, $Z_{a/2}$ = Z value at 95% confidence interval = 1.96, death rate = 50%, Hazard Ratio (HR) = 0.5, Survival probability = 0.5, the proportion of withdrawal = 0.15 [28].

The study population was preterm neonates on whom trophic feeding is more commonly practiced. Neonates that fulfill eligibility criteria were recruited from respective study hospitals. From neonates who were twin or triplet, only one of them was included by lottery method at each spot. Finally, 210 neonates were selected by using simple random sampling technique considering admission records as sampling frame.

## Measurements and variables

Time to initiate trophic feeding was the outcome variable of this study. Socio-demographic characteristics (Gestational age, birth weight, age of the mother, residence and educational status), neonatal related factors (First minutes APGAR score, fifth minutes APGAR score, meconium passage, sucking reflex, continuous positive air pressure, perinatal asphyxia, hemodynamic instabilities, birth defect, obstructions, respiratory distress syndrome, meconium aspiration syndrome, blood transfusion and Phototherapy) maternal related factors (sero-status of the mother, hypertensive disorder of pregnancy, mode of delivery, birth type, place of delivery, parity, postpartum hemorrhage and diabetes mellitus) and health service related factor (frequency of order revision) were the study's independent variables.

## Operational definitions

**Early feeding:** neonates start trophic feeding within 24 hours of birth [26,29].

   **Delayed feeding:** neonates start trophic feeding after 24 hours of birth [29].

   **Survival time:** the length of time in hours followed starting from birth to the first trophic feeding

   **Event:** the neonates who had started first trophic feeding within the follow-up period.

   **Censored:** neonates who died left against medical advice, transferred or referred before starting trophic feeding, or not started at end follow-up

   **Follow up time:** time from birth to the first seven days of life.

   **Hemodynamic instabilities:** Blood group and RH incompatibility, anemia, polycythemia, bleeding disorders, blood glucose disturbances [30].

   **Trophic feeding:** The first minimal enteral feeding to prime the gut regardless of method or volume [29].

## Data collection tools and procedure

Data was collected using a semi-structured pretested English version questionnaire and extraction checklist through face to face interview and chart review. The content of the questionnaire includes neonatal and maternal socio-demographic variables (gestational age, birth weight, age of the mother, residence and educational status), neonatal (first and fifth minutes APGAR score, meconium passage, sucking reflex, continuous positive air pressure, perinatal asphyxia, hemodynamic instabilities, birth defect, obstructions, respiratory distress syndrome, meconium aspiration syndrome, blood transfusion, phototherapy) and maternal related factors

(sero-status of the mother, hypertensive disorder during pregnancy, mode of delivery, birth type, place of delivery, parity, postpartum hemorrhage and diabetes mellitus) and health service-related factors (frequency of order revision). The data extraction checklist and questionnaire were adapted from different related literature, books, and guidelines [17,26,31–39].

Besides the principal investigator, six nurses working at NICU, two from each respective hospital, as a data collector, and three nurses as a supervisor were participated throughout the data collection process and data extraction was also done by data collectors.

Baseline data were obtained soon after admission, and the rest of the data were obtained every day in the follow-up period. Both supervisors and the principal investigator checked the completeness and consistency of the data on the daily basis.

## Data quality control

The questionnaire and the checklist was prepared in English and translated to Amharic which is the local and working language in the study area and back to English by language experts to maintain consistency.

Prior to data collection, one-day training was given for data collectors and supervisors on the study objectives, data collection instruments, techniques, and producers. A pretest was done on 11 neonates (5%) at Debre Markos specialized hospital and necessary amendments were done based on the pre-test findings. The consistency and completeness of data were checked by the principal investigator and supervisors on daily basis.

## Data processing and analysis

The data were checked for completeness, coded, and entered into Epi-data version 3.1; and exported to Stata/SE 14.0 for data cleaning and analysis.

Continuous data were reported with a mean (standard deviation) and median (interquartile range). The data with categorical nature was described with frequency and proportion. The outcomes of study participants were dichotomized into (code '1') as a failure (starting trophic feeding) and (code '0') as a censor. Some continuous variables were categorized for ease of analysis and otherwise used as continuous. The variance inflation factor (VIF) and correlation matrix were used to assess multi-collinearity.

The Kaplan Meier survival curve was used to estimate survival time, and a log-rank test was used to compare the survival curves of categorical variables. The necessary assumption of the Cox-proportional hazard regression model was checked using the Schoenfeld residual test, the graphical methods, and the presence of a time-dependent covariate. The overall model adequacy and fineness were assessed using the Cox-Snell residuals and global fit test, respectively. The Log likelihood ratio was used to select the final variables of the model and also bi-variable Cox-regression was computed for each predictor variable and a P-value of <0.25 was used as a cut-off point to enter variables to multi-variable Cox-regression. The variables were selected through backward stepwise procedures. The confounding effect was minimized using proper inclusion and exclusion criteria and a multi-variable analysis.

The result of the final model was expressed in terms of adjusted hazard ratio (AHR) with 95% confidence intervals. The significant association was declared with a p-value less than 0.05 in a multivariable Cox regression model. Finally, the result of this study is presented with tables, graphs, or text narrations.

## Ethics approval and consent to participate

The ethical approval letter was obtained from Debre Markos University research and ethical committee, college of medicine and health science (Ref.No/HSC/R/C /Ser/co/44/ 11/13). In

addition, letters of authorization was obtained from Debre Markos Specialized Hospital, Tibebe Ghion Specialized Hospital and Felege Hiwot Specialized Hospitals before contacting the participants. The participants were then fully briefed about the study's purpose and benefits and obtained informed written consent for both data collection and publication. Confidentiality was maintained through anonymity and privacy measures were taken to preserve the right of the participants throughout the research work including publication. Finally, the selected participants were asked about their willingness to join the study. Any study participant willing to engage in the study and those who wanted to stop an interview at any time were allowed to do so. This study was conducted in accordance with the Declaration of Helsinki.

## Results

### Neonatal and maternal socio-demographic characteristics

A total of 278 preterm neonates were hospitalized to these units in this prospective follow-up study conducted from October 1 to November 30, 2020. Of these 68 were ruled out. This left with a total of 210 preterm neonates with the mean gestational age of 33 (±3 standard deviation) completed weeks and a minimum of 28 to a maximum of 36 weeks. About 191 (90.95%) neonates were started TF and 19 (9.1%) were censored due to death.

Among neonates included in the study, all of them were low birth weight with a mean weight of 1549.9 (±353.5 standard deviation) grams with a minimum of 850 grams to a maximum of 2400 grams. Concerning weight for gestational age, about 157 (74.8%) was appropriate for their gestational age. The mean age of mothers was 27.4 (±5.05 standard deviation) years with a minimum of 18 years and a maximum of 40 years. Among mothers interviewed, greater than half (60.9%) were residing in a rural area (Table 1).

### Maternal and neonatal related factors

The mean APGAR-score of neonates at the first and fifth minute was 6 (±1.96 standard deviation) and 7.4 (±1.29 standard deviation), respectively. In this study, most (86.38%) of neonates were passed their first meconium before starting TF. Among neonates included in the study, 54.76% were referred from other health care facilities to the study hospitals. More than half 142 (67.62%) of them were born out of the study hospitals with spontaneous vaginal delivery.

**Table 1. Socio-demographic characteristics of the mothers and neonates admitted in NICU of DMCSH, FHCSH, and TGCSH comprehensive specialized hospitals, Amhara, Northwest Ethiopia, 2020 (N = 210).**

| Variables | Categories | Frequency | Percentage |
|---|---|---|---|
| Maternal education | Not attend formal education | 79 | 37.6 |
| | Primary | 47 | 22.4 |
| | Secondary | 47 | 22.4 |
| | Secondary &above | 37 | 17.6 |
| Gestational age in week | <34 weeks | 124 | 59.05 |
| | ≥ 34 weeks | 86 | 40.95 |
| Birth weight in gram | 850–1499 | 86 | 40.95 |
| | 1500–2499 | 124 | 59.05 |
| Maternal age | ≤19 | 11 | 5.2 |
| | 20–24 | 56 | 26.7 |
| | 25–29 | 65 | 31 |
| | 30–34 | 58 | 27.6 |
| | ≥35 | 20 | 9.5 |

**Table 2. Characteristics of neonates and their mothers 'admitted in NICU of DMCSH, FHCSH, and TGCSH comprehensive specialized hospitals, Amhara, Northwest Ethiopia, 2020.**

| Variables | Frequency | Percentage |
|---|---|---|
| APGAR score at 1st minute | | |
| <7 score | 110 | 52.38 |
| ≥7 score | 100 | 47.62 |
| APGAR at 5th minute | | |
| <7 score | 46 | 21.90 |
| ≥7 score | 164 | 78.10 |
| Sustained sucking reflex of the neonate until starting TF | | |
| No | 137 | 65.24 |
| Yes | 73 | 34.76 |
| Hemodynamic instability | | |
| No | 170 | 80.95 |
| Yes | 40 | 19.05 |
| Order revision frequency | | |
| At least every 24 hours | 92 | 43.81 |
| Greater than 24 hours | 118 | 56.19 |
| Respiratory distress syndrome | | |
| No | 88 | 41.90 |
| Yes | 122 | 58.10 |
| Maternal education | Not attend formal education | 79 |
| | Primary | 47 |
| | Secondary | 47 |
| | Secondary &above | 37 |
| Gestational age in week | <34 weeks | 124 |
| | ≥ 34 weeks | 86 |
| Birth weight in gram | 850–1499 | 86 |
| | 1500–2499 | 124 |
| Maternal age | ≤19 | 11 |
| | 20–24 | 56 |
| | 25–29 | 65 |
| | 30–34 | 58 |
| | ≥35 | 20 |

Almost half (54.76%) of mothers were primiparous and 110 (52.38%) of their current birth was a singleton (Table 2).

## Survival status of neonates on time to initiate TF

A total of 210 neonates to mother pairs were followed for 10136 person-hours of risk time. The minimum and maximum time of follow-up was 5 and 141 hours, respectively. The cumulative probabilities of starting TF by the end of 24, 48, and 72 hours were 22%, 53.8%, and 72.74%, respectively. The overall incidence of starting TF was 2 per 100 (95% CI: 1.6, 2.2) person-hours. The incidence rate that preterm neonates start TF was 5.67, 2.51, 1.53, and 0.94 per 100 person-hours in the first 24, 48, 72, and greater than 72 hours after birth, respectively. The median follow-up time was 38 (IQR: 25–70) hours. The median survival time was 42 (IQR: 26–72) (95% CI: 36, 48) hours (Table 3).

In addition to the overall survival estimate, the survival experience of neonates with different categorical variables was executed to compare the status of TF between or across groups.

**Table 3.** Survival probabilities to start TF among neonates admitted in NICU of DMCSH, FHCSH, and TGCSH comprehensive specialized hospitals, Amhara, Northwest Ethiopia, 2020.

| Time interval | Beginning total | Start feeding | Censored | Cumulative survival probability | 95% CI |
|---|---|---|---|---|---|
| 0 to 24 | 210 | 41 | 6 | 0.8019 | (0.74, 0.85) |
| 24 to 48 | 163 | 71 | 10 | 0.4416 | (0.37, 0.52) |
| 48 to 72 | 82 | 31 | 2 | 0.2726 | (0.21, 0.34) |
| 72to 96 | 49 | 25 | 0 | 0.1335 | (0.09, 0.19) |
| 96 to 120 | 24 | 13 | 1 | 0.0597 | (0.03, 0.11) |
| 120 to 144 | 10 | 10 | 0 | 0.0000 | . . |

For instance, there was a significant difference in median survival status which was 48 (IQR: 32–74, CI: 42–57) and 30 (IQR: 24–41, CI: 20–56) hours among neonates with gestational age of less than 34 weeks and 34 and more weeks, respectively. Similarly, neonates who had APGAR score of less than seven did not start TF until the median time of 55 (IQR: 30–90, CI: 42–67) hours and those with APGAR score of seven and above were not started TF until the median time of 32 (IQR: 20–52, CI: 28–41) hours. Neonates born outside of the hospital were did not start TF 53 (IQR: 30–90, CI: 42–69) hours, whereas neonates born inside the study hospitals were did not start TF until the median survival time of 32 (IQR: 22–49, CI: 28–42) hours (Table 3). Further analysis comparing the median time to initiate TF among neonates on presence of respiratory distress syndrome showed that there was a significant difference in median starting time, 53 (IQR: 33–82, CI: 45–65) hours and 27 (IQR: 16–47, CI: 22–32) hours among those with and with no respiratory distress syndrome. The statistical significance of the difference in the survival experience of TF was checked with a log-rank test (p<0.05) (Table 4).

From graphs below, the first graph shows the overall survival curves to initiate TF among followed neonates (**Fig 1**).

As indicated in Fig 2 the survival probability of initiating TF was longer in neonates with a birth weight of <1500 grams compared to those with ≥1500 grams (**Fig 2**).

**Table 4.** The group-specific median time of starting TF among neonates admitted in NICU of DMCSH, FHCSH, and TGCSH, Amhara, Northwest Ethiopia from October to November /2020 (n = 210).

| Predictors | Category | Median survival time in hours (95% CI) | Log-rank test p-Value |
|---|---|---|---|
| Respiratory distress syndrome | No | 27 (21.42, 32.58) | <0.0001 |
| | yes | 53 (44.80, 61.20) | |
| Perinatal asphyxia | No | 37 (31.24, 42.76) | 0.002 |
| | Yes | 65 (45.32, 84.68) | |
| Hemodynamic instability | No | 37 (31.36, 42.65) | <0.0001 |
| | Yes | 96 (83.62, 108.38) | |
| Gestational age | <34 weeks | 48 (42.07, 53.93) | <0.0001 |
| | ≥34 weeks | 30 (24.71, 35.29) | |
| 1st minute APGAR score | ≥7 score | 32 (25.32, 38.68) | <0.0001 |
| | <7 score | 55 (42.19, 67.81) | |
| Place of delivery | in-born | 32 (27.50, 36.50) | <0.0001 |
| | out-born | 53 (38.19, 67.81) | |
| Mode of delivery | SVD | 36 (30.21, 41.79) | 0.002 |
| | Cesarean section | 60 (40.16, 79.85) | |

SVD- spontaneous vaginal delivery.

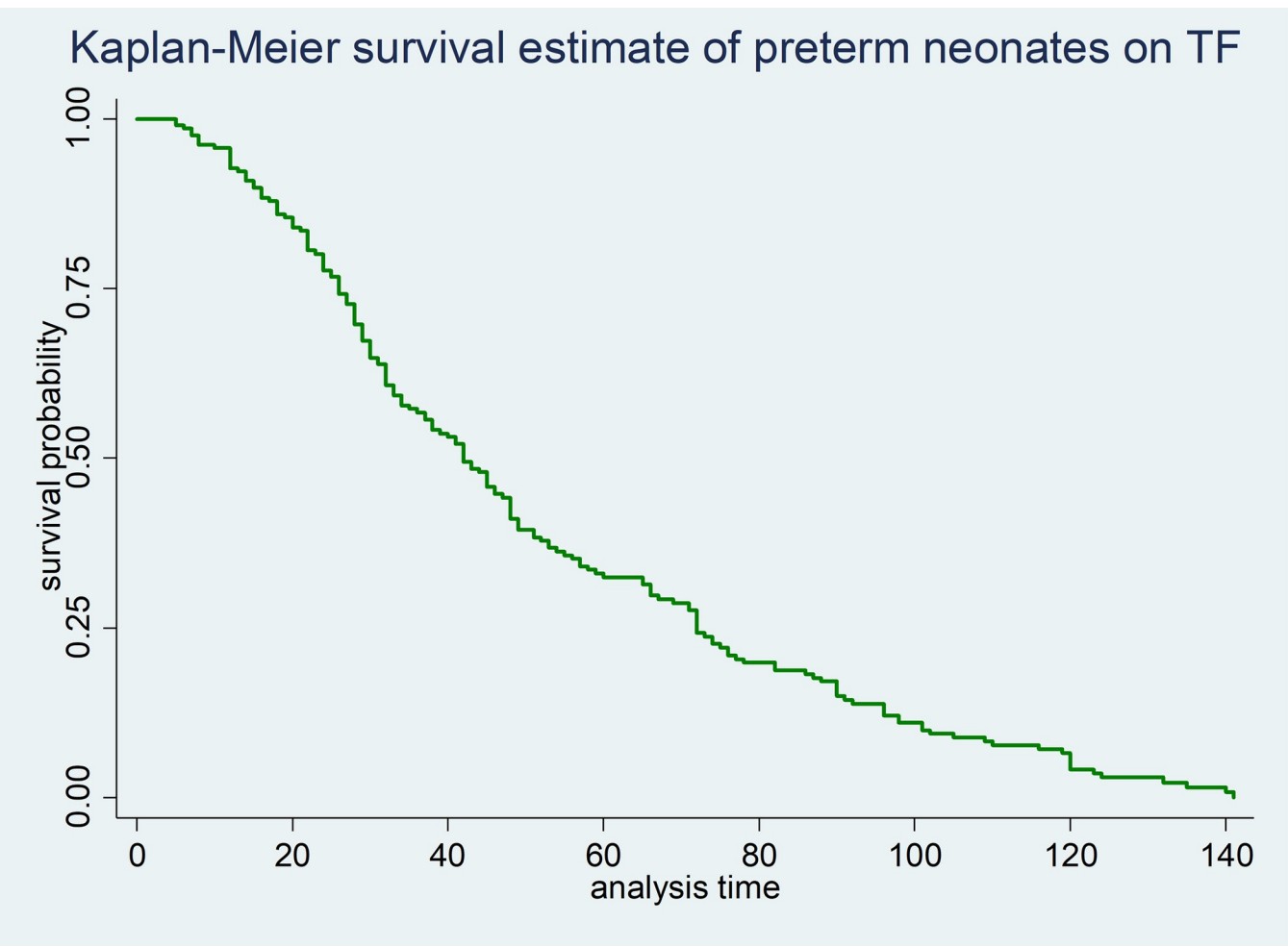

**Fig 1. The Kaplan-Meier survival estimates of time to initiate TF among neonates admitted to NICU of DMCSH, FHCSH, and TGCSH, Amhara, Northwest Ethiopia from October to November/2020 (n = 210).**

## Cox-proportional hazard assumption

The assumptions of Cox proportional hazard of all predictor variables were checked with graphical method of log-log plot curves and Kaplan–Meier curves among different groups of categorical predictors. The Schoenfeld residual test and time-varying covariate were used to test Cox proportional hazard assumptions statistically.

## Predictors of time to initiate TF

After bi-variable cox regression, diabetes mellitus, gestation, educational status, residence, age of the mother, meconium aspiration syndrome, meconium passage, sucking reflex, obstructions, blood transfusion, sero-status of the mother, and birth defect were not entered into multi-variable analysis because of p-value > 0.25. Then, Multivariable analysis was executed for gestational age, birth weight, Weight for gestational age, first minutes APGAR-score, fifth minutes APGAR-score, continuous positive air pressure, phototherapy, perinatal asphyxia, hemodynamic instabilities, respiratory distress syndrome, frequency of order revision, maternal hypertensive disorder of pregnancy, parity, mode and place of delivery. Finally, seven

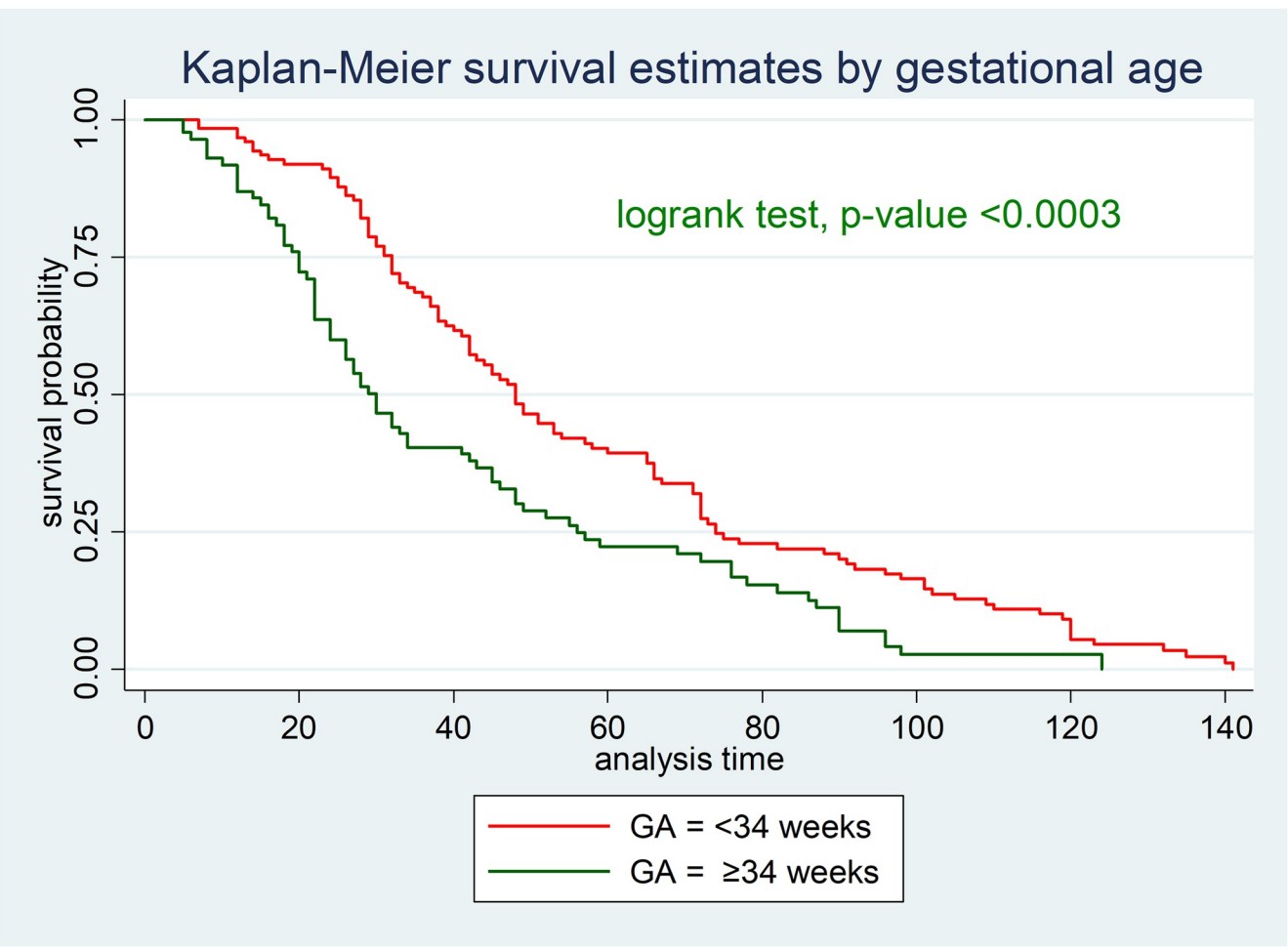

**Fig 2. Kaplan-Meier survival estimate of starting TF based on the gestational age of neonates admitted in NICU of DMCSH, FHCSH, and TGCSH, Amhara, Northwest Ethiopia from October to November/2020 (n = 210).**

variables were identified as statistically significant independent predictors (95% CI) for time to initiate TF (Table 5).

Multi-variable cox regression analysis revealed that the first minute APGAR-score, gestational age (less than 34 weeks), respiratory distress syndrome, perinatal asphyxia, small weight for gestational age, hemodynamic instability, and place of birth and mode of delivery were statistically significant variables of this study.

Accordingly, the hazard of Starting TF among neonates scored below seven APGAR at the first minute were 40% less likely as compared to neonates with seven and above score (AHR: 0.6, 95% CI: 0.44, 0.82). The hazard of starting TF among neonates born with less than 34 weeks of gestation was 31% less likely compared to those born with 34 and above weeks of gestation (AHR: 0.69, 95% CI: 0.5, 0.94) (Table 4).

The hazard of starting TF was 52% less likely among neonates who had respiratory distress syndrome compared to their counterparts (AHR: 0.5, 95% CI: 0.36, 0.68) (Fig 3).

The hazard of starting TF among neonates who have hemodynamic instability was 67% less likely compared to neonates with no these problems (AHR: 0.37, 95% CI: 0.24, 0.56). The

**Table 5. The final model containing predictors of time to initiate TF among neonates who were admitted to NICU of DMCSH, FHCSH and TGCSH, Amhara, Northwest Ethiopia from October to November /2020 (n = 210).**

| Predictor variables | Categories | Starting TF | | CHR (95% CI) | AHR (95% CI) |
|---|---|---|---|---|---|
| | | Yes | No | | |
| First minute APGAR score | <7 score | 99 | 11 | 0.48 (0.35, 0.64) | 0.6 (0.44, 0.82)* |
| | ≥7 score | 92 | 8 | 1 | 1 |
| Gestational age | <34 weeks | 113 | 11 | 0.59 (0.44, 0.78) | 0.69 (0.5, 0.94)* |
| | ≥34 weeks | 78 | 8 | 1 | 1 |
| Respiratory distress syndrome | Yes | 112 | 10 | 0.47 (0.35, 0.63) | 0.5 (0.36, 0.68)** |
| | No | 79 | 9 | 1 | 1 |
| Perinatal asphyxia | Yes | 45 | 5 | 0.59 (0.42, 0.82) | 0.63 (0.44, 0.89)* |
| | No | 146 | 14 | 1 | 1 |
| Weight for gestational age | SGA | 47 | 6 | 0.7 (0.54, 0.97) | 0.74 (0.52, 1.04) |
| | AGA | 144 | 13 | 1 | |
| Hemodynamic instability | Yes | 39 | 1 | 0.26 (0.17, 0.38) | 0.37 (0.24, 0.57)** |
| | No | 152 | 18 | 1 | 1 |
| Place of birth | Out-born | 107 | 8 | 0.47 (0.36, 0.65) | 0.54 (0.39, 0.74)** |
| | In-born | 84 | 11 | 1 | 1 |
| Mode of delivery | CS | 63 | 5 | 0.62 (0.46, 0.84) | 0.63 (0.44, 89)* |
| | SVD | 128 | 14 | 1 | 1 |

Note

* indicates p-value <0.05

** indicates p-value ≤0.001, both * and ** indicates statistically significant variables in the multi-variable analysis, CHR-crude hazard ratio, AHR-adjusted hazard ratio, CI-confidence interval, SVD- spontaneous vaginal delivery, CS-cesarean section.

hazard among neonates diagnosed with perinatal asphyxia was 36% less likely to start TF compared with their counterparts (AHR: 0.63, 95% CI: 0.44, 0.89).

In addition to these predictors, the hazard of giving delivery with cesarean section was 31% less likely to start TF than that of spontaneous vaginal delivery (AHR: 0.63, 95% CI: 0.44, 89) (Table 4). Neonates, who were not born within the study hospitals, were 47% less likely to start TF than those born within the study hospitals (AHR: 0.54, 95% CI: 0.39, 0.74).

## Statistical model and assumptions

The value of log likelihood ratio was used to select different models by adding and removing variables entered in to multivariable regression. Then, a model with variables resulting maximum likelihood ratio was taken as final model for interpretation.

The overall adequacy of the final fitted model was checked by the Cox-Snell residuals were estimated based on the Kaplan–Meier estimated survivor function. This graphical plot of the cumulative hazard versus cox-Snell residuals showed an approximate straight line with slope one and indicates the model fits the data. And in addition to the Cox-Snell residuals graph, the final model was tested statistically for its fitness by using the global goodness of fit test which was decided as adequate if its p-value is greater than 0.05. Based on this, the global goodness of fit test for this final model was p-value = 0.59 (Fig 4).

## Discussion

This study aimed to estimate the time to initiate TF and its predictors among preterm neonates admitted in the study hospitals within the study period. In this study, the incidence of starting

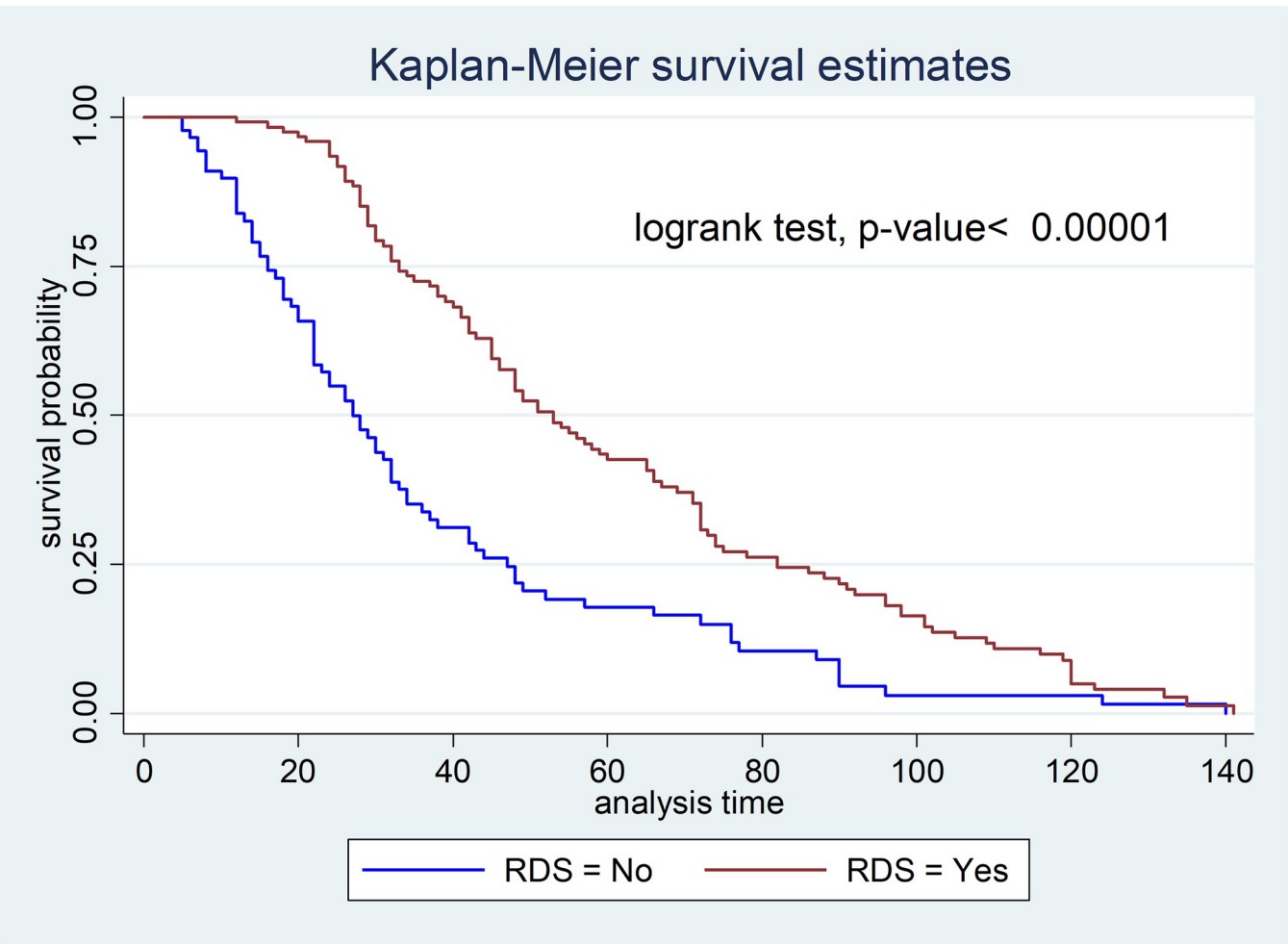

**Fig 3. Kaplan-Meier survival estimate of starting TF based on respiratory distress syndrome among neonates admitted in NICU of DMCSH, FHCSH, and TGCSH, Amhara, Northwest Ethiopia from October to November/2020 (n = 210).**

TF was 2 per 100 person-hours of risk time. At the end of follow-up, 90.9% of neonates (95% CI: 86.2%, 94.2%) were started TF. The rest of neonates (9.1%) were not started TF as a result of death just before starting TF. Among those who started TF, only 52 (24.76%) of preterm neonates started within the first 24 hours of birth. This indicates that only a small proportion of preterm neonates started TF within the first 24 hours of birth. This finding is lower as compared to an observational study in Tuscany (74.1%), a cohort study in Iran (36%), and New Zealand (60%) [21,40]. The finding of this study on the proportion of neonates who were started TF within 48 hours was consistent with the finding of a cohort study conducted on preterm neonates in NICU of the Islamic Republic of Iran (63.2%) [21], but higher than the finding of Nigerian Special Care Baby Unit (40%) [41]; and lower than finding of a study conducted in Uganda rural hospitals (80%) [26] and New Zealand (80%) [37].

This prospective follow-up study, the median time to start TF was 42 (IQR: 26–72) hours which is higher than the finding of a retrospective study in New Zealand (24 hours) [42]. The difference might be due to the study setup, study design (prospective versus retrospective), sample size (210 versus 647), and socio-demographic variations, and differences in regional variation in neonatal management protocols (Ethiopia versus New Zealand).

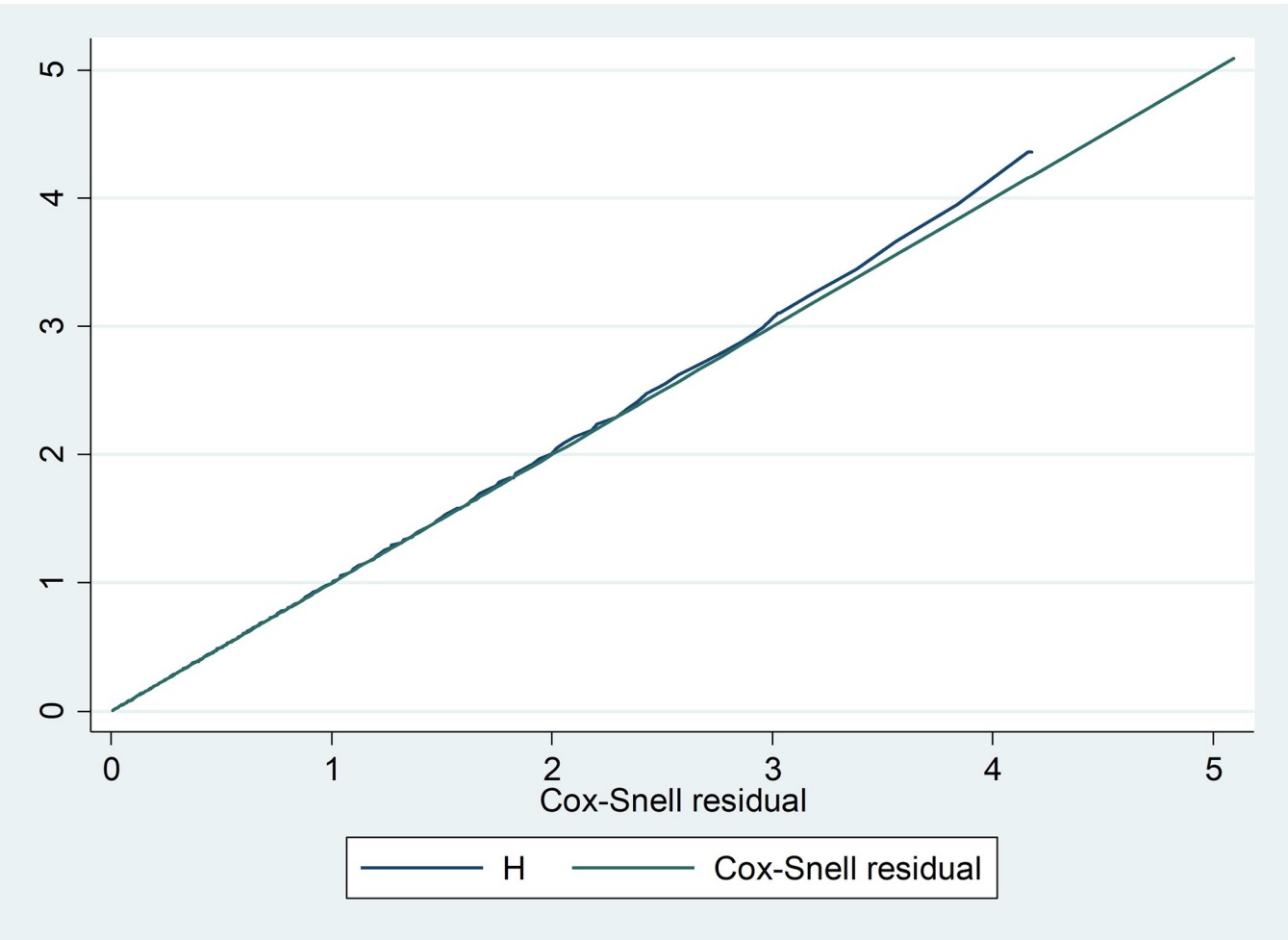

**Fig 4. Cox-Snell residual graph, based on the Kaplan–Snell estimated survivor function, to test the overall adequacy of the Cox proportional hazard model of time to initiate TF and its predictors among neonates admitted to NICU of DMCSH, FHCSH, and TGCSH, Amhara, Northwest Ethiopia, 2020 (N = 210).**

Multi variable cox regression of this study revealed that gestational age of below 34 weeks, APGAR-score of less than seven, out-born delivery, cesarean delivery, respiratory distress syndrome, perinatal asphyxia, and hemodynamic instability were predictors of delay in starting of trophic feeding.

The hazard of starting TF among neonates born less than 34 weeks of gestation was less likely compared to those born 34 and above weeks of gestation. This might be due to differences in physiological maturity among these groups of neonates on whom necrotizing enterocolitis and feeding intolerance are common in the former group of neonates. This finding is supported by a web-based survey on tertiary NICUs in different countries that revealed that the lower gestational age was the reason for the delayed starting of TF [43]. Another survey research done in the United States and Canada also identified lower gestational age as the primary predictor of TF [44]. This is also a reason to delay initiation of TF among neonates with a gestational age of <31 weeks. Additionally, a survey done on feeding practice among 98 Spanish neonatal units revealed that the smaller GA was the main predictor for delayed initiation of TF [39]. This is also supported with a prospective follow up study conducted in

teaching university hospitals in Ethiopia that showed the time neonates kept NPO was increased with lower gestational age [23].

The first minute APGAR-score was another statistically significant predictor. The hazard of starting TF among neonates scored below seven APGAR at the first minute was 40% less likely as compared to neonates with a seven and above score. This finding might be because that first-minute APGAR-score is the signal of intrauterine hypoxia which is an indication of decreased blood flow to GIT and increased risk of NEC. Likewise, the hazard among neonates diagnosed with perinatal asphyxia was 37% less likely to start TF compared with their counterparts. This might be due to professionals' fear of necrotizing enterocolitis and feeding intolerance secondary to intrauterine hypoxia and decreased blood flow to GIT. This finding is supported with an observational study in Spain, reported that perinatal asphyxia was the reason for the delay in 88% of the neonate [39]. But, contrary to this, the Ethiopian national NICU guideline recommended that TF can be started for asphyxiated neonates if the neonate is passing meconium, clear gastric content, normo-active bowel sound [29]. Additionally, a guideline on preterm neonates recommended that birth asphyxia was not a contraindication to initiate TF [45]. Another statistically significant predictor was hemodynamic instability. With this regard, the hazard of starting TF among neonates who have hemodynamic instability was 67% less likely compared to neonates with the absence of these problems. This indicates neonates who had hemodynamic instability were more likely have delayed initiation of TF than those who had no such problems. The presence of these groups of problems such as blood group and RH incompatibility, anemia, polycythemia, and bleeding disorders may affect clinicians' decision to initiate TF due to fear of perceived feeding intolerance and necrotizing enterocolitis. This finding is supported by an observational study in Spain that reported hemodynamic instabilities were the reason for the delay to start TF in 100% of the neonates [39]. Another study conducted in regional referral hospitals in Tanzania showed hemodynamic instability and its perceived risk of necrotizing enterocolitis was the reason for the delayed initiation of TF [46].

Furthermore, the hazard of starting TF in neonate with respiratory distress syndrome was 50% less likely among neonates compared with their counterparts. This may be because the presence of respiratory distress might be aggravated with pressure from the abdominal cavity that is further complicated by physiological instability with bradycardia and desaturation events, which have been associated with an increased risk of aspiration during feeding [47]. This may influence professionals' decision to delay starting TF. Contrary to this finding, some guidelines across the globe recommend a minimal EF can be initiated in neonates with respiratory distress syndrome within the first 24 hours of life [45].

Place of delivery was also found to be statistically significant predictors of time to initiate TF. Neonates born out of the study hospitals were 48% less likely to start TF than those born within the study hospitals. However, there is a lack of evidence to support the relationship between place of birth and time to initiate trophic feeding; the possible explanation might be due to transportation delay.

The hazard of giving delivery with a cesarean section was 37% less likely to start TF as compared with those delivered via spontaneous vaginal delivery. Mothers, who gave birth with cesarean section, might not bring expressed milk upon request by the clinician. In contrast to this finding, research reports revealed that mode of delivery had no statistically significant effect on time to initiate enteral feeding [48]. This difference might be due to the difference in sample size which was 729 in the previous study and 210 in this study, difference in the study design, and general socio-demographic characteristics of the study population.

## Limitation of the study

The sample size was calculated with Stata package because of an absence of previous related literature that reported essential parameters used for calculation.

## Conclusion

There was a considerable delay to initiate trophic feeding. The time of starting trophic feeding was delayed in the study hospitals. After adjustment for confounding, APGAR-score of less than seven, gestational age of below 34 weeks, being out-born, cesarean delivery, presence of respiratory distress syndrome, perinatal asphyxia, and hemodynamic instability were predictors found to hinder the time of starting trophic feeding. Therefore, health care institutions providing neonatal intensive care services shall work on these mentioned predictors to shorten the initiation of trophic feeding and to reduce its complications related to the delay.

## Acknowledgments

The authors would like to acknowledge the Debre Markos University of Gondar, College of Health Sciences for support of this research project. The authors also extend their special thanks to both data collectors and supervisors.

## Author Contributions

**Conceptualization:** Daniel Adimasu Kebede, Yilikal Tafere, Tewodros Eshete, Ermias Abebaw, Mekonen Adimasu.

**Data curation:** Daniel Adimasu Kebede.

**Formal analysis:** Daniel Adimasu Kebede, Tewodros Eshete, Mekonen Adimasu, Bekalu Endalew.

**Investigation:** Daniel Adimasu Kebede.

**Methodology:** Daniel Adimasu Kebede, Yilikal Tafere, Tewodros Eshete, Ermias Abebaw, Mekonen Adimasu, Bekalu Endalew.

**Project administration:** Daniel Adimasu Kebede.

**Software:** Daniel Adimasu Kebede, Tewodros Eshete, Mekonen Adimasu.

**Supervision:** Daniel Adimasu Kebede, Yilikal Tafere.

**Validation:** Daniel Adimasu Kebede.

**Visualization:** Daniel Adimasu Kebede.

**Writing – original draft:** Daniel Adimasu Kebede, Ermias Abebaw.

**Writing – review & editing:** Daniel Adimasu Kebede, Ermias Abebaw.

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
