## [Decision Letter · Decision Letter 0]

26 May 2022

PONE-D-21-32465

The Time To Initiate Trophic Feeding And Its Predictors Among Preterm Neonate Admitted To Neonatal Intensive Care Unit, Multicenter study, Northwest Ethiopia, 2020.

PLOS ONE

Dear Dr. Belay,

Thank you for submitting your manuscript to PLOS ONE. After careful consideration, we feel that it has merit but does not fully meet PLOS ONE’s publication criteria as it currently stands. Therefore, we invite you to submit a revised version of the manuscript that addresses the points raised during the review process.

Your manuscript has been assessed by two expert reviewers, whose comments are appended to this letter. The reviewers have raised several concerns about the presentation of the methodology, details of the statistical analysis and aspects of the conclusions. Please ensure you respond to each of the points carefully in your response to reviewers and revise your manuscript accordingly.

We look forward to receiving your revised manuscript.

Kind regards,

Joseph Donlan

Editorial Office

PLOS ONE

**Journal requirements:**

3. Please upload a copy of Figure 5, to which you refer in your text on page 20. If the figure is no longer to be included as part of the submission please remove all reference to it within the text.

**Reviewers' comments:**

Reviewer's Responses to Questions

**Comments to the Author**

1. Is the manuscript technically sound, and do the data support the conclusions?

Reviewer #1: Yes

Reviewer #2: Yes

2. Has the statistical analysis been performed appropriately and rigorously? 

Reviewer #1: Yes

Reviewer #2: Yes

3. Have the authors made all data underlying the findings in their manuscript fully available?

Reviewer #1: Yes

Reviewer #2: Yes

4. Is the manuscript presented in an intelligible fashion and written in standard English?

Reviewer #1: Yes

Reviewer #2: Yes

5. Review Comments to the Author

Reviewer #1: Your manuscript entitled "The Time To Initiate Trophic Feeding And Its Predictors Among Preterm Neonate

Admitted To Neonatal Intensive Care Unit, Multicenter study, Northwest Ethiopia, 2020" has the potential to be published...

The standard of English language was suitable. It described a technically sound piece of scientific research with data that supports the conclusions.

Reviewer #2: Comment 1: Title and aim of the study should be revisited. In Abstract (Line 40), it was mentioned that aim of the study to assess the time to initiate the TF, In background (Line 116): The aim of the study to estimate the time of initiate the TF. The Title should suggest predictors of early initiation of TF or survival of neonates who have been initiated TF early?

Background Section:

Comment 2: Line 111: The initiation, mode, and rate of advancement of TF remained a topic of controversy. Revise the sentence with a more suitable word other than "controversy."

Comment 3: Line 135: Kindly revise or define the word " health professional mixes."

Comment 4: Line 137 and 139 Extend the short name when authors use it the first time. kangaroo mother care (KMC) and continuous positive airway pressure (CPAP)

Comment 5: Use the proper way to write the date: Line 139. Write November 30, 2020 instead of November 30/2020.

Methodology Section:

The methodology needs revision.

Comment 6: Revisit the flow: Line 140 & 141 should be placed after inclusion and exclusion criteria or during the description - sample selection. Here, it creates duplications.

Comment 7: Inclusion and Exclusion Criteria: Does the study include or exclude neonates given TF at previously admitted hospitals? Does the study include the neonate who had hemodynamic instability or any other complications after initiating the TF?

Comment 8: Revisit line 158." Neonates that don't fulfill eligibility criteria were recruited from respective study hospitals": Why did the study recruit the non-eligible study samples?

Comment 9: Line 160: Check the English grammar and describe: was it a simple random sampling method or a different randomization method?

Comment 10: Revisit the Data collection method: Mention the variables included in the checklists, brief on the contents of the semi-structured questionnaire and who has undertaken the data collection for both.

Comment 11: The study also assigned the NICU nurses as data collectors and supervisors. Kindly explain - How the data collection process was not biased despite assigning NICU working nurses & supervisors for data collection? Who checked the completeness of the data? (Authors or Supervisors?) What steps had been taken to avoid it?

Comment 12: Line 179: "The outcomes of study participants were dichotomized into (code '1') as a failure and (code '0') as a censor." What was the outcome assigned to the study participants?

Result Section: Required to maintain the flow of the results based on objectives and planned analysis.

Comment 13: Check the grammar in lines 212 (278 preterms) and 213 (of these, 48..)

Comment 14: Line 228: Referred to study hospitals (outborn) or referred to other centers from study hospitals.? Please clarify.

Comment 15: Line 234: The cumulative probabilities of starting or the cumulative proportions?

Comment 16: Line 240: Describe the survival estimates to build the flow for model of predictors.

Comment 17: Defined all categorical variables in methodology. (line 243-244)

Comments 18: Revisit the line 255, 256

Comment 19:

median survival time of the neonates would have higher impact on the data variables. Here, I recommend to reframe the results. In Table 3; Mention that the survival (time to initiate the feeding) time (hours)

The variables in Cox hazard and Kaplan-Meier survival model, the most appropriate subset of these predictors are to be selected in the multivariable model based on their contribution to the maximized log partial likelihood of the model (−2LL).

The interpretations based on further analysis and translate the findings. The other important step is considering variables that are nonsignificant at univariable analysis but it could be confounders. The effect of adding each of the variables should be checked with −2LL(). It was undertaken here. but this should be outlined in the methodology and in result section with title of the statical model.

At the final step, the Wald test is to be used to assess the significance of reasonable and possible interactions if possible.

Discussion :

Remove duplication of the results. Reduce number of words.

Comment 20: Line 300. To assess or to estimate.

Comment 21: Line 301, English Grammar- study hospitals

Comment 22: Line 323: Reframe the sentence

Comment: 23 Discussion should also mention the neonates were not started TF and reasons.

Conclusion:

Outline the specific recommendations revealed out from the study. in the conclusion section, it is not clear to the reader what are the specific recommendations for the hospitals. Clear 'Action points' need to be described accordingly.

6. PLOS authors have the option to publish the peer review history of their article (what does this mean?). If published, this will include your full peer review and any attached files.

Reviewer #1: **Yes: **Shooka Mohammadi

Reviewer #2: **Yes: **Dr Harsh Shah

---

## [Author Response · Author response to Decision Letter 0]

29 Jun 2022

Point by point response 

Title: The Time to Initiate Trophic Feeding and Its Predictors among Preterm Neonate Admitted to Neonatal Intensive Care Unit, Multicenter study, Northwest Ethiopia.

 Review Comments to the Author

Comment : Reviewer #1: Your manuscript entitled "The Time To Initiate Trophic Feeding And Its Predictors Among Preterm Neonate Admitted To Neonatal Intensive Care Unit, Multicenter study, Northwest Ethiopia, 2020" has the potential to be published...

The standard of English language was suitable. It described a technically sound piece of scientific research with data that supports the conclusions

Response: Thank you and comments are accepted. 

 Comment: Reviewer #2: Comment 1: Title and aim of the study should be revisited. In Abstract (Line 40), it was mentioned that aim of the study to assess the time to initiate the TF, In background (Line 116): The aim of the study to estimate the time of initiate the TF. The Title should suggest predictors of early initiation of TF or survival of neonates who have been initiated TF early?

Response: We appreciate this insight but the aim of this study was to estimate the time of initiation to TF because there was no baseline study indicating the median time of TF initiation. Hence the first study shall be estimating the time of initiation to TF then after the above raised issue will proceed. Therefore, we didn’t investigate neonates who have started TF early before admission as indicated in the exclusion criteria.

Background Section:

Comment 2: Line 111: The initiation, mode, and rate of advancement of TF remained a topic of controversy. Revise the sentence with a more suitable word other than "controversy."

Response: We appreciate this intention and we revised this sentence. See line 91 in the revised manuscript.

Comment 3: Line 135: Kindly revise or define the word “health professional mixes."

Response: we have revised. See line 115 in the revised manuscript.

Comment 4: Line 137 and 139 Extend the short name when authors use it the first time. kangaroo mother care (KMC) and continuous positive airway pressure (CPAP)

Response: Thank you, we now corrected it.

Comment 5: Use the proper way to write the date: Line 139. Write November 30, 2020 instead of November 30/2020.

Response: It is corrected on the revised manuscript.

Methodology Section:

The methodology needs revision.

Comment 6: Revisit the flow: Line 140 & 141 should be placed after inclusion and exclusion criteria or during the description - sample selection. Here, it creates duplications.

Response: We tried to re-arrange the whole methodology part as per the comment given.

Comment 7: Question1: Inclusion and Exclusion Criteria: Does the study include or exclude neonates given TF at previously admitted hospitals? 

Response: We excluded neonates who were started TF at referring hospitals. See line 126-127 of the revised manuscript. 

Question2: Does the study include the neonate who had hemodynamic instability or any other complications after initiating the TF?

Response: No, because the aim of this study was assessing time to starting trophic feeding; follow up was until the event (starting trophic feeding) developed. So we don’t have any data about the study participants after the follow up period. 

Comment 8: Revisit line 158." Neonates that don't fulfill eligibility criteria were recruited from respective study hospitals": Why did the study recruit the non-eligible study samples?

Response: Sorry, it was type error and corrected. See line 137 in the revised manuscript.

Comment 9: Line 160: Check the English grammar and describe: was it a simple random sampling method or a different randomization method?

Response: We checked the grammar and the sampling method we used was simple random sampling technique by using the admission registration as sampling frame. See line 139-140 in the revised manuscript. 

Comment 10: Revisit the Data collection method: Mention the variables included in the checklists, brief on the contents of the semi-structured questionnaire and who has undertaken the data collection for both.

Response: We revised this section accordingly. See line 168-176 &180 of the revised manuscript. We also tried to mention the independent variables and the questionnaire contents at the “measurement and variables section” of our revised manuscript.

Comment 11: The study also assigned the NICU nurses as data collectors and supervisors. Question1: Kindly explain - How the data collection process was not biased despite assigning NICU working nurses & supervisors for data collection? 

Response: Before data collection there was one day training for both data collectors and supervisors who were working at NICU of all study areas on the study objective, data collection tools, procedures and necessary methodological setups. Next to this our study objective was to estimate the time to initiate trophic feeding, this issue does not measure the performance of the NICU nurses and their responsibility is giving the service while the mothers came to their room. The primary responsibility of early linking for trophic feeding is for Midwives who are working at the delivery room. So firstly, we give training on the purpose of the study; next the principal investigator checked the data on daily basis randomly by taking the medical registration number of the study participants and lastly the topic/ issue is not as such sensitive for the NICU nurses. Therefore, we assured that the data were not biased by data collectors and supervisors as we tried to justify in the above. 

Question 2 : Who checked the completeness of the data? (Authors or Supervisors?) What steps had been taken to avoid it?

Response: The data was checked by both the assigned supervisor and the principal investigator. The steps are mentioned in the response of question one. See line 182-183 of the revised manuscript. 

Comment 12: Line 179: "The outcomes of study participants were dichotomized into (code '1') as a failure and (code '0') as a censor." What was the outcome assigned to the study participants?

Response: The outcome variable of this study was time to initiate trophic feeding and failure coded as”1”were neonates who started trophic feeding during the follow up period. See line 198-199. 

Result Section: Required to maintain the flow of the results based on objectives and planned analysis.

Response: We tried to maintain the flow of the result accordingly. 

Comment 13: Check the grammar in lines 212 (278 preterms) and 213 (of these, 48..)

Response: We checked and amend accordingly. See line 232-233 of the revised manuscript. 

Comment 14: Line 228: Referred to study hospitals (out born) or referred to other centers from study hospitals? Please clarify.

Response: Those neonates referred from other health facilities to study hospitals and we corrected in the revised manuscript. See line 250-251.

Comment 15: Line 234: The cumulative probabilities of starting or the cumulative proportions?

Response: It was cumulative probability. Because survival probability is calculated as the number of subjects surviving divided by the number of patients at risk and the reported numbers were done so. https://www.ncbi.nlm.nih.gov/pmc/articles/PMC3059453/

Comment 16: Line 240: Describe the survival estimates to build the flow for model of predictors

Response: We described the survival estimates of each predictor in the revised manuscript. See line 271-284.

Comment 17: Defined all categorical variables in methodology (line 243-244).

Response: We corrected it. Please See line 141-151 of the revised manuscript. 

Comments 18: Revisit the line 255, 256

Response: We appreciate this insight and we amended it accordingly. See line 348-351 of the revised manuscript.

Comment 19: median survival time of the neonates would have higher impact on the data variables. Here, I recommend to reframe the results. In Table 3; Mention that the survival (time to initiate the feeding) time (hours). The variables in Cox hazard and Kaplan-Meier survival model, the most appropriate subset of these predictors are to be selected in the multivariable model based on their contribution to the maximized log partial likelihood of the model (−2LL).

The interpretations based on further analysis and translate the findings. The other important step is considering variables that are non-significant at uni-variable analysis but it could be confounders. The effect of adding each of the variables should be checked with −2LL (). It was undertaken here. But this should be outlined in the methodology and in result section with title of the statical model. At the final step, the Wald test is to be used to assess the significance of reasonable and possible interactions if possible.

Response: We tried to correct it based on your suggestion. See line 107-108 of the methodology section and line 348-351 of the result section of the revised manuscript. 

Discussion :

Remove duplication of the results. Reduce number of words.

Response: We saw the whole discussion part thoroughly and acted accordingly. 

Comment 20: Line 300. To assess or to estimate.

Response: To estimate 

Comment 21: Line 301, English Grammar- study hospitals

Response: We corrected it. See line 367

Comment 22: Line 323: Reframe the sentence

Response: We reframed it. Please see line 390-391. 

Comment: 23 Discussion should also mention the neonates were not started TF and reasons.

Response: We tried to add on it.

Conclusion:

Outline the specific recommendations revealed out from the study. In the conclusion section, it is not clear to the reader what are the specific recommendations for the hospitals. Clear 'Action points' need to be described accordingly

Response: We wrote the specific recommendation based on our findings accordingly.

---

## [Decision Letter · Decision Letter 1]

22 Jul 2022

The Time To Initiate Trophic Feeding And Its Predictors Among Preterm Neonate Admitted To Neonatal Intensive Care Unit, Multicenter study, Northwest Ethiopia.

PONE-D-21-32465R1

Dear Dr. Belay,

We’re pleased to inform you that your manuscript has been judged scientifically suitable for publication and will be formally accepted for publication once it meets all outstanding technical requirements.

Kind regards,

Harsh Shah, MD

Guest Editor

PLOS ONE

Additional Editor Comments (optional):

Reviewers' comments:

Reviewer's Responses to Questions

**Comments to the Author**

1. If the authors have adequately addressed your comments raised in a previous round of review and you feel that this manuscript is now acceptable for publication, you may indicate that here to bypass the “Comments to the Author” section, enter your conflict of interest statement in the “Confidential to Editor” section, and submit your "Accept" recommendation.

Reviewer #1: All comments have been addressed

2. Is the manuscript technically sound, and do the data support the conclusions?

Reviewer #1: Yes

3. Has the statistical analysis been performed appropriately and rigorously? 

Reviewer #1: Yes

4. Have the authors made all data underlying the findings in their manuscript fully available?

Reviewer #1: Yes

5. Is the manuscript presented in an intelligible fashion and written in standard English?

Reviewer #1: Yes

6. Review Comments to the Author

Reviewer #1: our manuscript entitled "The Time To Initiate Trophic Feeding And Its Predictors Among Preterm Neonate

Admitted To Neonatal Intensive Care Unit, Multicenter study, Northwest Ethiopia, 2020" has the potential to be published...

It described a technically sound piece of scientific research with data that supports the conclusions.

7. PLOS authors have the option to publish the peer review history of their article (what does this mean?). If published, this will include your full peer review and any attached files.

Reviewer #1: **Yes: **Dr. Shooka Mohammadi

---

## [Editor Report · Acceptance letter]

2 Aug 2022

PONE-D-21-32465R1 

The Time to Initiate Trophic Feeding and Its Predictors among Preterm Neonate Admitted to Neonatal Intensive Care Unit, Multicenter study, Northwest Ethiopia 

Dear Dr. Abebaw:

I'm pleased to inform you that your manuscript has been deemed suitable for publication in PLOS ONE. Congratulations! Your manuscript is now with our production department. 

Kind regards, 

on behalf of

Dr. Harsh Shah 

Guest Editor

PLOS ONE